# Extensively Hydrolyzed Hypoallergenic Infant Formula with Retained T Cell Reactivity

**DOI:** 10.3390/nu15010111

**Published:** 2022-12-26

**Authors:** Raphaela Freidl, Victoria Garib, Birgit Linhart, Elisabeth M. Haberl, Isabelle Mader, Zsolt Szépfalusi, Klara Schmidthaler, Nikos Douladiris, Alexander Pampura, Evgeniy Varlamov, Tatiana Lepeshkova, Evgeny Beltyukov, Veronika Naumova, Styliani Taka, Dina Nosova, Olga Guliashko, Michael Kundi, Alina Kiyamova, Stefani Katsamaki, Rudolf Valenta

**Affiliations:** 1Center for Pathophysiology, Infectiology and Immunology, Institute of Pathophysiology and Allergy Research, Medical University of Vienna, A-1090 Vienna, Austria; 2International Center of Molecular Allergology, Ministry of Innovation Development, Tashkent 100174, Uzbekistan; 3HiPP GmbH & Co. Vertrieb KG, 85276 Pfaffenhofen, Germany; 4Department of Pediatrics and Adolescent Medicine, Division of Pediatric Pulmonology, Allergy and Endocrinology, Comprehensive Center of Pediatrics, Medical University Vienna, A-1090 Vienna, Austria; 5Allergy Department, 2nd Pediatric Clinic, National & Kapodistrian University of Athens, 11527 Athens, Greece; 6Department of Allergology and Clinical Immunology, Research and Clinical Institute for Pediatrics Named after Yuri Veltischev at the Pirogov Russian National Research Medical University of the Russian Ministry of Health, 117997 Moscow, Russia; 7Department of Faculty Therapy, Endocrinology, Allergology and Immunology, Ural State Medical University, 620014 Ekaterinburg, Russia; 8Allergy Department, UNIMED Laboratories, 119049 Moscow, Russia; 9Department for Environmental Heath, Center for Public Health, Medical University of Vienna, A-1090 Vienna, Austria; 10NRC Institute of Immunology FMBA of Russia, 119049 Moscow, Russia; 11Laboratory of Immunopathology, Department of Clinical Immunology and Allergy, Sechenov First Moscow State Medical University, 119049 Moscow, Russia; 12Karl Landsteiner University for Health Sciences, 3500 Krems, Austria

**Keywords:** allergy, cow’s milk allergy, allergen, hypoallergenic infant formula

## Abstract

Background: Immunoglobulin E (IgE)-mediated cow’s milk allergy (CMA) can be life-threatening and affects up to 3% of children. Hypoallergenic infant formulas based on hydrolyzed cow’s milk protein are increasingly considered for therapy and prevention of cow’s milk allergy. The aim of this study was to investigate the allergenic activity and ability to induce T cell and cytokine responses of an infant formula based on extensively hydrolyzed cow’s milk protein (whey) (eHF, extensively hydrolyzed formula) supplemented with Galactooligosaccharides (GOS) and *Limosilactobacillus fermentum* CECT5716 (LF) to determine its suitability for treatment and prevention of CMA. Methods: eHF and standard protein formula based on intact cow’s milk proteins (iPF) with or without Galactooligosaccharide (GOS) and *Limosilactobacillus fermentum* CECT5716 (LF) were investigated with allergen-specific antibodies and tested for IgE reactivity and allergenic activity in basophil degranulation assays with sera from cow’s milk (CM)-allergic infants/children. Their ability to stimulate T cell proliferation and cytokine secretion in cultured peripheral blood mononuclear cells (PBMC) from CM-allergic infants and children was studied with a FACS-based carboxyfluorescein diacetate succinimidyl ester (CFSE) dilution assay and xMAP Luminex fluorescent bead-based technology, respectively. Results: An eHF supplemented with GOS and LF exhibiting almost no IgE reactivity and allergenic activity was identified. This eHF induced significantly lower inflammatory cytokine secretion as compared to an intact protein-based infant formula but retained T cell reactivity. Conclusions: Due to strongly reduced allergenic activity and induction of inflammatory cytokine secretion but retained T cell reactivity, the identified eHF may be used for treatment and prevention of CMA by induction of specific T cell tolerance.

## 1. Introduction

Immunoglobulin E (IgE)-mediated cow’s milk allergy (CMA) affects approximately 3% of children, and manifestations usually occur early in children who receive cow’s milk in their diet early in life [1,2]. Patients with an IgE sensitization to cow’s milk may suffer from a broad spectrum of allergic symptoms that may affect the gastrointestinal tract as well as other organs like the skin, the respiratory tract and the cardiovascular system (e.g., life-threatening anaphylaxis) [3]. 

Interestingly, certain subjects with IgE sensitization to cow’s milk may also remain asymptomatic [4]. The severity of allergic symptoms in subjects with IgE sensitization to cow’s milk depends on several factors. For example, subjects with high levels of cow’s milk-specific IgE antibodies were reported to suffer from more severe and systemic symptoms as compared to subjects with lower IgE levels [5]. 

Cow’s milk contains several different allergen molecules that differ regarding their concentrations, allergenic activity and resistance to digestion and heating [6]. Accordingly, individuals with high levels of IgE antibodies against digestion-resistant and heat-stable milk allergens such as caseins may suffer from more severe symptoms. By contrast, subjects with an IgE sensitization to bovine serum albumin, which occurs in rather low levels in cow’s milk and is easily digested, often have no symptoms of cow’s milk allergy [7]. Additionally, other factors such as epithelial barrier function and the presence of neutralizing IgA antibodies that bind harmful antigens or allergen-specific IgG antibodies that block IgE binding to the allergens may influence the extent and quality of cow’s milk-related allergic symptoms [8]. 

IgE-mediated allergic sensitization is the major risk factor for severe and life-threatening anaphylactic reactions to cow’s milk. It is, therefore, important to determine if an individual carries IgE antibodies against cow’s milk allergens or suffers from other forms of intolerance such as lactose intolerance, which is very common [3]. For other forms of immunologically mediated hypersensitivity reactions to cow’s milk, there are currently no reliable diagnostic tests available that are based on the measurement of cow’s milk-specific antibodies, other than IgE or cellular tests, and, therefore, such tests are not recommended [3,9]. 

In the first months after birth, breastfeeding is the ideal nutrition. For patients with confirmed IgE sensitization to cow’s milk, standard treatment recommendations are: (i) cow’s milk avoidance by strict diet, or (ii) consumption of extensively hydrolyzed hypoallergenic infant formulas or amino acid-based infant formulas [10,11]. For children 4 years and older with persistent cow’s milk allergy, EAACI guidelines recommend allergen immunotherapy (AIT), although no uniform AIT protocols have been established yet [12]. Hypoallergenic formulas are especially important in early life when breastfeeding is not possible and are a suitable substitute for infant feeding provided the formula is hypoallergenic and safe [13]. This is usually the case for extensively hydrolyzed and amino acid-based formulas due to the destruction of IgE epitopes. 

Another property, especially of partially hydrolyzed, hypoallergenic infant formulas, is the possibility of inducing immunological tolerance against cow’s milk proteins, which requires the presence of cow’s milk allergen-derived intact T cell epitopes to induce either clonal deletion or anergy in specific T cells or regulatory T cell responses [8]. Hypoallergenic formulas with such properties could, therefore, not only be used for treatment of established cow’s milk allergy but also for prevention of the development of cow’s milk allergy, not only by avoiding the induction of IgE sensitization but also by the induction of specific tolerance [14]. 

However, it has been reported that the degree of hydrolysis differently affects the presence of IgE and T cell epitopes in formulas, and it is, therefore, necessary to characterize formulas regarding their immunological properties [15]. 

Whether the induction of cow’s milk allergen-specific tolerance is possible is a controversial issue because certain studies performed in animal models [16] suggest that the induction of tolerance is not likely [17]. Another recent study using an extensively hydrolyzed whey fraction indicates that tolerance induction can be achieved [18]. Here, we studied such an extensively hydrolyzed whey fraction regarding its allergenic properties and effects on T cell proliferation and cytokine production in the blood of cow’s milk-allergic infants and children. We identified an extensively hydrolyzed whey fraction that had greatly reduced allergenic activity and induced significantly less production of inflammatory cytokines but at the same time retained allergen-specific epitopes recognized by human T cells. Our findings suggest that this formula combines the features of reduced allergenic activity required for treatment and maintains T cell epitopes for possible tolerance induction. 

## 2. Materials and Methods

### 2.1. Cow’s Milk-Allergic Patients, Sera, PBMC Samples

A demographic and clinical description of individuals analyzed in this study is provided in Table 1. In total, 59 individuals were analyzed in the study, of whom 49 had reported symptoms upon cow’s milk consumption (Table 1). Ten control individuals were included, of whom seven had other allergies but not cow’s milk allergy (NCMA 48, 49, 50, 51, 53, 55 and 56, Table 1), and three were non-allergic subjects (NA1–NA3, Table 1). 

The patients were from the Department of Pediatrics and Adolescent Medicine, Medical University Vienna, Vienna, Austria, the Allergy Department, 2nd Pediatric Clinic University of Athens, Greece, the Department of Allergology and Clinical Immunology, Research and Clinical Institute for Pediatrics named after Yuri Veltischev at the Pirogov Russian National Research Medical University, Moscow, Russia and from the Department of Faculty Therapy, Endocrinology, Allergology and Immunology, Ural State Medical University, Ekaterinburg, Russia. Written informed consent was obtained from their parents, and blood sampling/analysis was performed under pseudonymized conditions with approval by the local ethics committees in Greece and Russia (N10/15.12.2017, N12/20.12.2017) and the ethics committee of the Medical University of Vienna, Austria (EK1641/2014). The analyses of patients’ sera and blood samples were performed in an anonymized manner. 

The diagnosis of CMA was based on the presence of clinical symptoms of CMA that could be unambiguously attributed to cow’s milk consumption, and/or a positive skin prick test reaction to cow’s milk, results from an open food challenge and/or detection of specific IgE to CM allergens as measured by ImmunoCAP or ImmunoCAP ISAC (Thermo Fisher Scientific, Uppsala, Sweden). The cow’s milk-related allergic symptoms of patients (mean age: 4.9 years; median age: 4.1 years) are summarized in Table 1 and included anaphylaxis as graded by Sampson [19], atopic dermatitis and/or gastrointestinal symptoms such as abdominal pain, vomiting, diarrhea, and blood in stool. Most of the CMA patients had allergic symptoms to other food allergen sources such as egg, peanut, wheat, nuts, soy, cereals, fish, and caviar (Table 1). 

### 2.2. Allergens, Materials, Infant Formulas, Antibodies, SDS-PAGE

Purified natural cow’s milk allergens, skim milk powder and human serum albumin (HSA) were purchased from Sigma Aldrich (St. Louis, Missouri, #9045-23-2, 9048-46-8, 9000-71-9, 9000-71-9, 9000-71-9, BCR685, 70024-90-7). Extensively hydrolyzed formula (eHF) with Galactooligosaccharides (GOS), intact protein formula (iPF) (i.e., cow´s milk protein that has not been hydrolyzed) with and without Galactooligosaccharides (GOS), *Limosilactobacillus fermentum* CECT5716 (LF), and extensively hydrolyzed whey protein (raw material Peptigen^®^ IF-3080, eH_raw) were provided by HiPP GmbH & Co. Vertrieb KG (Pfaffenhofen, Germany) and Arla Foods Ingredients (Videbæk, Denmark), respectively (Table 2). In more detail, HiPP HA infant formula was manufactured from Peptigen^®^ IF-3080 (supplied by Arla Foods Ingredients, Videbæk, Denmark). Peptigen^®^ IF-3080 is an extensive whey protein hydrolysate suitable as the sole protein source in infant formula. It consists of short-chain peptides, obtained by a controlled enzymatic degradation of whey proteins. Hydrolysis is performed with food grade enzymes that are heat-inactivated upon termination of hydrolysis. Subsequently, the hydrolysate is filtered by ultrafiltration in order to remove larger peptides and aggregates thereof. Peptigen^®^ IF-3080 has a degree of hydrolysis of up to 30%. The degree of hydrolysis is defined as the percentage of peptide bonds cleaved by hydrolysis and determined according to Adler-Nissen and Nielsen et al. [20,21].

Freeze-dried *Limosilactobacillus fermentum* CECT5716 (LF) was added without further culture or passage to certain formulas/materials in a final concentration of 1.5 × 10^6^ CFU/1 mL (CFU, colony forming unit) before experiments or tested as such. This concentration corresponded to that of the marketed HiPP infant formula and meets the amount of bacteria observed in human milk samples [22]. 

Customized in-vitro digestion simulating full passage through the oral, gastric and small intestinal phase was performed in a SHIME^®^ model by ProDigest BV (Ghent, Belgium). In vitro digested samples were consistently labeled dig_ (Table 2, samples 8–12).

CM allergen-specific rabbit anti had been raised against the purified CM allergens, and normal rabbit serum (nrs) was used as control [6].

Proteins, materials and formulas were analyzed by 12% SDS-PAGE by loading aliquots of 1 µg natural CM allergens (α-casein, β-casein, κ-casein, α-lactalbumin, β-lactoglobulin) or 20 µL of infant formulas/materials (eHF + GOS, iPF + GOS, eH_raw, iPF). The SDS-PAGE was then stained with Coomassie-Brilliant Blue R250 or blotted onto nitrocellulose and reacted with CM-allergen-specific antisera as previously described [6]. The concentrations of proteins and peptides in the investigated materials were determined by bicinchoninic acid assays for proteins or peptides (BCA Protein and Peptide Assay Kits, Thermo Scientific, # 23225, 23275).

### 2.3. Immune Dot Blot

Immune dot blots were performed as previously described [14]. Briefly, aliquots of 1 µg of materials (i.e., extensively hydrolyzed infant formulas, digested infant formulas, intact cow’s milk protein-based formulas, probiotic *L. fermentum* CECT5716, HSA or skim milk) were dotted on nitrocellulose membranes (Whatman Protran; Sigma-Aldrich) and dried. After blocking, stripes were incubated with CM allergen-specific rabbit antisera that had been raised against the purified CM allergens [6], and bound allergen-specific rabbit antibodies were detected with ^125^I-labeled goat anti-rabbit antibodies (Perkin Elmer, Waltham, MA, USA) and visualized by autoradiography. Likewise, stripes were incubated with sera from cow’s milk allergic patients or, for control purposes, with sera from non-allergic subjects (diluted 1:10) overnight (typically for 15–16 h) at 4 °C, before bound IgE antibodies were detected with 1:10 diluted ^125^I-labeled anti-human IgE antibodies and visualized by autoradiography [14]. Buffer control (BC) without addition of primary antibodies (i.e., allergen-specific rabbit antisera or human sera) was performed with PBS containing 0.1% Tween for experiments performed with allergen-specific rabbit antibodies and with PBS containing 0.5% Tween and 0.5% BSA for experiments performed with human sera. 

Signals obtained by autoradiography were obtained at identical exposure times to allow a comparison of signal intensities, which were arbitrarily compared and termed as lacking, weak, distinct = medium or strong.

### 2.4. Basophil Degranulation Assays

Basophil degranulation assays were performed with rat basophil leukemia (RBL) cells expressing the human high-affinity receptor for IgE as previously described [23]. 

In brief, RBL cells cultured in duplicates expressing the human FcεRI α/β/γ subunits were loaded with 1:10 diluted sera from 20 CM-sensitized patients from whom sufficient amounts of serum were available (#1, 3, 5, 6, 14, 16, 17, 21, 22, 25, 27, 28, 29, 35, 36, 37, 38, 46, 47, 54, Table 1) overnight, and degranulation was induced by adding antigens in a concentration of 10 ng/mL. The concentration 10 ng/mL was determined in pilot experiments to be representative for the increasing part of the bell-shaped curve of basophil degranulation by testing concentrations of 100, 10, 1 or 0.1 ng/mL (Appendix A). A concentration of 10 ng/mL of different allergen molecules was also identified in earlier work to represent the increasing part of the bell-shaped basophil degranulation curve using the RBL cell line [23]. The release of β-hexosaminidase supernatants was analyzed by incubating culture supernatants with 80 μmol/L 4-methylumbelliferyl -N-acetyl-β-D-glucosaminide (Sigma-Aldrich) in citrate buffer (0.1 mol/L, pH 4.5) for 1 h at 37 °C. The reaction was stopped by addition of 100 μL of glycine buffer (0.2 mol/L glycine and 0.2 mol/L NaCl, pH 10.7), and fluorescence was measured at an extinction wavelength of 360 nm to the emission wavelength of 465 nm by using a fluorescence microplate reader (CYTO FLUOR 2350; Millipore, Billerica, MA, USA). Results are reported as the percentage of total β-hexosaminidase released after complete cell lysis achieved by addition of 10% Triton X-100 (Merck, Darmstadt, Germany).

Results represent the average of duplicates with deviations of less than 10% and background (i.e., incubation with HSA was subtracted). The mean percentages +/− SEM were calculated for the group of 20 patients, and statistical analysis was performed as indicated in the section “Statistical analysis”. 

### 2.5. FACS-Based Analysis of the Proliferation of CD4^+^ and CD8^+^ T Cells in Response to Antigens by CFSE Dilution

Peripheral blood mononuclear cells (PBMCs) were isolated from heparinized blood samples of nine patients/subjects (Table 1: patients/subjects 48–56) using Ficoll density gradient centrifugation (Amersham Biosciences, Uppsala, Sweden). Aliquots of 200,000 PBMCs in 200 µL were stained with carboxyfluorescein diacetate succinimidyl ester (CFSE) dye, which distributes among dividing cells [24]. Gating was performed on CD3^+^ and CD4^+^ and CD3^+^ and CD8^+^ cells, respectively [25]. The cells were incubated in triplicate for 7 days at 37 °C in a humidified atmosphere with 5% CO_2_ in 96-well plates (Nunclone; Nalgen Nunc International, Roskilde, Denmark) in Ultra Culture medium (BioWhittaker, Rockland, ME, USA) supplemented with 2 mM L-glutamine (Gibco, Carlsbad, CA, USA) and 50 µM beta-mercaptoethanol (Gibco) to prevent free radical build-up, and 0.1 mg gentamicin per 500 mL (Gibco) in the presence of allergens Bos d 4 (natural α-lactalbumin, nALA), Bos d 5 (natural β-lactoglobulin, nBLG), Bos d 8 (natural casein, ncasein), and infant formulas (10 µg protein/well) in triplicates. As positive control, T cell activator (1 μL/well) (Dynabeads Human T-Activator CD3/CD28; Thermo Fisher Scientific/Invitrogen) was used. Medium alone and 10 µg protein/well HSA served as a negative control [25]. For each of the 20 patients, cultivation with the different antigens was performed in triplicate. For each of the triplicates, the median of the medium-only wells was subtracted. For further statistical analysis, the mean of the three medium-corrected proliferation values was used. 

### 2.6. Measurement of Cytokine Levels

Cytokine levels (IL-1b, IL-2, IL-4, IL-5, IL-6, IL-10, IL-12, IL-13, IL-17, IFN-g, GM-CSF; interleukin, IL; interferon-gamma, IFN-g, granulocyte-macrophage colony-stimulating factor, GM-CSF) were measured in supernatants collected from PBMC cultures at day 7 of culture using xMAP Luminex fluorescent bead-based technology according to the manufacturer’s instructions (R&D Systems, Wiesbaden, Germany) [24]. The fluorescent signals were read on a Luminex 100 system (Luminex Corp., Austin, TX, USA). The limits of detection were 0.28 pg/mL for IL-1b, 1.7 pg/mL for IL-2, 0.2 pg/mL for IL-4, 5.6 pg/mL for IL-5, 0.45 pg/mL for IL-6, 0.81 pg/mL for IL-10, 2.01 pg/mL for IL-12, 0.36 pg/mL for IL-13, 2.74 pg/mL for IL-17, 1.67 pg/mL for IFN-g, and 0.43 pg/mL for GM-CSF. Results were analyzed as described for the analysis of T cell proliferations.

### 2.7. Statistical Analysis

Before evaluation, all variables were subjected to a distribution analysis of residuals after subtracting the estimated means. Except for results of the basophil degranulation assay that did not significantly deviate from a normal distribution, all variables were best fitted by a log-normal distribution and, therefore, log-transformed for subsequent statistical tests. For graphical presentation, means and ±SEM (standard error of mean) were back-transformed to express results on the original scale. Statistical comparisons were based on a generalized estimating equations (GEE) model with an unstructured correlation matrix to account for the fact that the same blood specimens were used for all formulas. Comparisons between formulas were restricted to pre-specified contrasts. For cytokines and T-cell proliferation, only eHF and iPF were compared. For the specific basophil degranulation formulas, eHF + GOS, iPF + GOS, eHF + GOS + LF, and iPF + GOS + LF were compared against skim milk, and eHF + GOS as well as iPF + GOS formulas with and without LF were compared by linear contrast with Bonferroni correction. The hypothesis test applied was Wald’s chi-square test. Correction was determined by the number of comparisons that were not orthogonal. 

All statistical tests were performed by Stata 13.0 (StataSoft). Figures were generated by Statistica 10.0 (StatSoft) (*) *p*-value < 0.05, (**) *p*-value < 0.01, (***) *p*-value < 0.001. 

## 3. Results

### 3.1. Extensively Hydrolyzed Infant Formulas Lack Intact Cow’s Milk Allergens

Figure 1A shows the analysis of the extensively hydrolyzed formula with GOS (eHF + GOS), intact protein formula (iPF + GOS, iPF) with and without Galactooligosaccharide (GOS), as well as of the extensively hydrolyzed whey protein (eH_raw) in comparison with purified natural CM allergens (α-casein, β-casein, κ-casein, α-lactalbumin, β-lactoglobulin) by SDS-PAGE and Coomassie Brilliant Blue staining. We found that neither the raw material (eH_raw), which builds the protein basis for the extensively hydrolyzed formula eHF + GOS, nor the extensively hydrolyzed formula eHF + GOS itself contained intact proteins. By contrast, bands corresponding to the caseins, alpha-lactalbumin and ß-lactoglobulin were found in the intact protein formula with and without GOS (iPF + GOS, iPF). 

After this analysis, we tested the different formulas, raw materials and purified cow’s milk allergens for reactivity with rabbit antisera raised against purified cow’s milk allergens (i.e., α-S1-casein, α-S2-casein, α-β-casein, α-κ-casein, α-lactalbumin, αβ-lactoglobulin, and α-lactoferrin) (Figure 1B,C). αS1/S2-casein, ß-casein, κ-casein, α-lactalbumin and ß-lactoglobulin but not lactoferrin were detected in skim milk and in undigested intact protein formula (iPF) with and without Galactooligosaccharides (GOS) (Figure 1C), but not in intact protein formulas that had been subjected to in vitro gastrointestinal digestion (dig_IPF + GOS, dig_iPF + GOS + LF, Figure 1B). None of the cow’s milk allergens were detected in the raw material (eH_raw) for the extensively hydrolyzed formula eHF + GOS, or in the extensively hydrolyzed formula eHF + GOS regardless of whether they were digested or not (Figure 1B). A weak signal was observed when the anti-ß-casein antiserum was tested with *Limosilactobacillus fermentum* CECT5716 (*LF*) (Figure 1C) or with formulas containing *Limosilactobacillus fermentum* CECT5716 (*LF*) (Figure 1B). Weak signals were observed when the undigested extensively hydrolyzed formulas were tested with antisera specific for kappa casein, alpha-lactalbumin and ß-lactoglobulin (Figure 1B). No signal was observed when samples were tested with normal rabbit serum or buffer without addition of rabbit antibodies (negative controls) (Figure 1B,C). No reaction of any of the antisera with HSA was observed (Figure 1B,C).

### 3.2. Extensively Hydrolyzed Formulas and In Vitro Digested Intact Protein Formulas Show Strongly Reduced IgE Reactivity

In the next set of experiments, we investigated the IgE reactivity of infant formulas, raw materials and skim milk using sera from cow’s milk-allergic patients and, for control purposes, with sera from non-allergic subjects (Table 1) (Figure 2A–D).

The majority of cow’s milk-allergic patients (i.e., patients 1, 2, 3, 5, 6, 8, 9, 10, 12, 14, 17, 18, 19, 21, 22, 24, 25, 26, 27, 28, 29, 35, 37, 39, 42, 44, 46, 47, 54) showed distinct = medium to strong IgE reactivity to dot-blotted intact milk (skim milk) and intact protein formulas (i.e., iPF, iPF + GOS, iPF + GOS + LF) (Figure 2C,D).

Some cow’s milk-allergic patients showed weak (i.e., patients 7, 13, 15, 16, 20, 31, 36, 40, 45) or no (i.e., patients 4, 11, 23, 30, 32, 33, 34, 38, 41, 43, 48–53, 55, 56) IgE reactivity to the aforementioned intact protein formulas (Figure 1C,D).

No IgE reactivity was observed for non-allergic subjects (NA1, NA2) or when buffer without addition of serum was tested with any of the dotted samples (Figure 1A–D). Sera that had shown distinct or strong IgE reactivity to intact protein formulas showed no or weak IgE reactivity to extensively hydrolyzed raw material, to undigested as well as to digested extensively hydrolyzed formulas, and digested intact protein formulas. Of note, patients 22 and 27 showed residual IgE reactivity to extensively hydrolyzed formulas but not to digested intact formulas, whereas patient 9 showed residual IgE reactivity to digested intact protein formulas but not to extensively hydrolyzed formulas (Figure 2A,B). No IgE reactivity to HSA or *Limosilactobacillus fermentum (LF)* was observed (Figure 2A–D).

### 3.3. Limosilactobacillus fermentum CECT5716 (LF)-Containing Extensively Hydrolyzed Formula Shows the Strongest Reduction of Allergenic Activity

In order to determine the allergenic activity of the tested formulas, basophil activation experiments were performed. For this purpose, we used rat basophil leukemia (RBL) cells that express the human high-affinity receptor for IgE and hence can be loaded with serum IgE from cow’s milk-allergic patients, and then allergen-specific and IgE-mediated degranulation can be induced by the addition of allergens. Allergen-induced and IgE-mediated basophil degranulation is dose-dependent but results in a bell-shaped activation curve because excess of allergen will result in a lower rate of IgE cross-linking. Therefore, we determined in a pilot experiment the dose, which yielded degranulation in the increasing part of the bell-shaped curve (Appendix A), and then performed basophil degranulation with this dose. Figure 3 displays the mean percentages of ß-hexosaminidase release for 20 cow’s milk-allergic patients who had shown IgE reactivity to cow’s milk allergens (i.e., skim milk) and extensively hydrolyzed and intact protein formulas with and without *Limosilactobacillus fermentum* CECT5716 (*LF*). The strongest basophil degranulation was obtained with skim milk, which was significantly stronger than that observed for the intact protein formula and the extensively hydrolyzed formula (Figure 3). ß-hexosaminidase release induced by the extensively hydrolyzed formula was significantly lower than that induced by the intact protein formula. Interestingly, the extensively hydrolyzed formula containing *Limosilactobacillus fermentum* CECT5716 (*LF*) (eHF + GOS + LF) showed almost no degranulation and induced significantly lower induction of basophil degranulation than the extensively hydrolyzed formula without *Limosilactobacillus fermentum* CECT5716 (*LF*) (eHF + GOS), whereas this effect of LF was not found for the intact protein formulas (Figure 3).

### 3.4. Extensively Hydrolyzed GOS-Containing Infant Formula Induces T Cell Proliferation Similarly to GOS-Containing Intact Protein Formula

After having studied the allergenic activity of infant formulas, we investigated their ability to induce specific T cell proliferation. Figure 4 shows the median percentages +/− SEM of proliferated CD4^+^ (left part of the figure) and CD8^+^ T cells (right part of figure) in PBMC from subjects 48–56 (Table 1) stimulated with a mix of natural caseins (ncasein), ß-lactoglobulin (nBLG), alpha-lactalbumin (nALA), extensively hydrolyzed infant formula or intact protein GOS-containing formula (eHF + GOS, iPF + GOS). The casein mix induced the strongest milk-specific CD4^+^ and CD8^+^ T cell proliferation, whereas the proliferation induced by whey allergens (ß-lactoglobulin, alpha-lactalbumin) was comparable to that induced by the extensively hydrolyzed formula. Specifically, there was no significant difference between the CD4^+^ and CD8^+^ T cell proliferation induced by extensively hydrolyzed and intact protein formulas (Figure 4, right parts). Stimulation with anti-CD3 and anti-CD28 antibodies (i.e., positive control) induced strong CD4^+^ and CD8^+^ T cell proliferation.

### 3.5. Extensively Hydrolyzed GOS-Containing Infant Formula Induces Lower Secretion of Inflammatory Cytokines than GOS-Containing Intact Protein Formula

We then investigated the ability of cow’s milk allergens and infant formulas to induce the secretion of cytokines in cultured PBMC samples from subjects 48–56 (Figure 5, Table 1). The extensively hydrolyzed GOS-containing infant formula (eHF + GOS) induced the secretion of lower levels of inflammatory cytokines than the intact GOS-containing formula (iPF + GOS), and this difference was significant for TNF-alpha, IL-1b, IL-2, IL-17, IL-4, IL-5, IL-13, IL-6 (data not shown) and GM-CSF (Figure 5). There was no significant difference regarding the levels of the tolerogenic cytokine IL-10 in cultures stimulated with eHF + GOS and iPF + GOS (Figure 5). We noted that except for MCP-1, whey allergens (i.e., ß-lactoglobulin, alpha-lactalbumin) induced higher levels of cytokines than caseins, although caseins had induced stronger T cell proliferation than whey allergens (Figure 4). Stimulation with anti-CD3 and anti-CD28 induced the production of each of the investigated cytokines in cultured PBMCs (Figure 5).

## 4. Discussion

Hydrolyzed formulas are obtained by enzymatic degradation of cow’s milk or fractions thereof [8]. Depending on the degree of hydrolysis, IgE epitopes and T cell epitopes can be differently affected [14]. The reduction of IgE reactivity serves the purpose of reducing the allergenic activity of the formula so that it can be consumed by cow’s milk-allergic infants and children without inducing IgE-mediated allergic inflammation. Accordingly, such formulas are suitable as nutrition for of cow’s milk-allergic infants and children, in particular when they cannot be breast-fed. The more a particular formula is hydrolyzed, the lower is its risk of inducing IgE-mediated allergic inflammation, but at the same time also, allergen-specific T cell epitopes are destroyed [10]. However, maintained T cell epitopes are important for the induction of preventive T cell-mediated tolerance. Basically three T cell-mediated mechanisms are thought to be involved in the preventive induction of specific immunological tolerance [8]. They comprise the induction of regulatory T cells (Tregs) usually at lower antigen doses [26] and clonal deletion or clonal anergy at higher antigen doses [27]. While studies have identified hydrolyzed formulas with reduced allergenic activity and reduced ability to induce specific T cell and cytokine responses in cow’s milk-allergic patients [13,14], studies performed in animals have yielded controversial results regarding whether hydrolyzed formulas can induce allergen-specific preventive tolerance. For example, one study demonstrated that the administration of extensive hydrolysates from caseins and *Lactobacillus rhamnosus* GG probiotic did not prevent cow’s milk protein allergy in a mouse model [17], whereas another study performed in rats showed that partially hydrolyzed whey had allergy-preventive capacity [18]. 

In this study, we investigated an extensively hydrolyzed infant formula and compared it with an intact protein formula regarding IgE reactivity, allergenic activity, and induction of specific T cell and cytokine responses in cow’s milk-allergic patients. We found that the eHF + GOS + LF lacked intact cow’s milk allergens as compared to iPF + GOS + LF and accordingly showed almost no IgE reactivity even when tested with sera from highly cow’s milk-allergic patients. We then studied the allergenic activity of the infant formulas using basophil degranulation experiments performed with rat basophilic leukemia cells that expressed the human FcεRI and hence could be loaded with serum IgE from cow’s milk-allergic patients. This experiment was of particular importance because it examines whether hydrolyzed allergens can cross-link IgE on effector cells and thus induce immediate allergic inflammation. The basophil activation test based on rat basophilic leukemia cells expressing the human FcεRI was used because these cells can be loaded with serum IgE from cow’s milk-allergic patients in a highly controlled manner, and the test is not influenced by the presence of allergen-specific IgG, as it occurs in whole blood samples and may affect basophil activation. Therefore, this assay seems to be better suited than a whole blood assay because the possible influence of blocking allergen-specific IgG on basophil activation can be excluded and the results truly reflect the allergenic activity of tested allergens/formulas. Furthermore, it would have been difficult to obtain fresh blood samples from a representative number of cow’s milk-allergic children. 

In these experiments, it was demonstrated that eHF + GOS and eHF + GOS + LF had significantly reduced allergenic activity as compared to iPF + GOS and iPF + GOS + LF, respectively. However, in these experiments, it turned out that the allergenic activity of eHF + GOS + LF was even significantly lower than that of eHF + GOS and thus almost completely abolished, whereas no significant difference was observed when iPF + GOS was compared with iPF + GOS + LF. This finding may be explained by a down-regulation of inflammatory responses by *Limosilactobacillus fermentum* CECT5716 that was attributed to a reduction of TLR2/TLR4 expression in a murine model of allergic asthma in a recent study [28]. This anti-inflammatory effect of *Limosilactobacillus fermentum* CECT 5716 may be more pronounced when inflammatory cell activation is already reduced due to already reduced IgE cross-linking. Regardless of what the specific mechanism behind the almost completely abolished allergenic activity of the eHF + GOS + LF formula is, it clearly identified this formula as the least allergenic formula among those investigated in our study, which should be useful for the treatment of cow’s milk-allergic infants/children. 

Usually, extensive hydrolysis destroys not only IgE epitopes but also T cell epitopes and, therefore, may render an extensively hydrolyzed formula not useful for the induction of T cell-mediated tolerance and hence the prevention of allergen-specific IgE sensitization. According to the manufacturer’s information, the degree of hydrolysis of the eHF was up to 30%, which was confirmed by our SDS-PAGE analysis showing that eHF + GOS and eH_raw did not contain any visible protein bands even below 10 kDa. Nevertheless, eHF + GOS induced specific proliferation of CD4^+^ and CD8^+^ T cells and thus seems to contain peptides that are in the size of 9aa or at least long enough to fit and activate MHC class I and MHC class II.

In fact, we observed that there was no significant difference regarding the induction of T cell proliferation between eHF and iPF, suggesting that most of the allergen-specific T cell epitopes are preserved in eHF and that, therefore, this infant formula may indeed be used for preventive tolerance induction, similarly as reported for an extensively hydrolyzed whey product in earlier animal experiments [18]. Moreover, eHF, despite preservation of allergen-specific T cell epitopes, induced significantly lower amounts of inflammatory cytokines (i.e., TNF alpha, IL-1ß, IL-2, IL-17, IL-4, IL-5, IL-13, IL-6 and GM-CSF) in PBMC cultures of CMA patients and thus induced also lower inflammatory responses in addition to reduced mast cell activation. Although the eHF + GOS and eHF + GOS + LF infant formulas showed strongly reduced allergenic activity in vitro, it must be borne in mind that the strength of allergic reactions may vary in patients. Accordingly, in vivo testing, for example by provocation testing, will be necessary to confirm our results. It may be considered as a limitation of our study that we did not investigate IgG and IgG_1_ subclass reactivity of the formulas due to the fact that international guidelines recommend against IgG testing [29]. Should clinical relevance of IgG_1_ reactivity be confirmed in the future, the testing of formulas for IgG_1_ reactivity may be considered. 

In summary, from a preclinical point of view our study identified eHF as a hypoallergenic formula and gives a first promising hint that it may be safely used for the treatment of cow’s milk allergy in already-sensitized infants and children. The formula hydrolysate also may be used for the specific primary prevention of cow’s milk allergy in not-yet sensitized children. Noteworthy, from a clinical and preclinical point of view both possibilities must be investigated in additional clinical trials. Despite the fact that efficacy in regard to preventing and treating CMA still needs to be demonstrated in infants, a recent clinical trial demonstrated nutritional safety and suitability in terms of growth (EFSA opinion: https://www.efsa.europa.eu/de/efsajournal/pub/7141; assessed on 10 December 2022). It is thus obvious that more extensive clinical data are needed to support our promising preclinical results.

## Figures and Tables

**Figure 1 nutrients-15-00111-f001:**
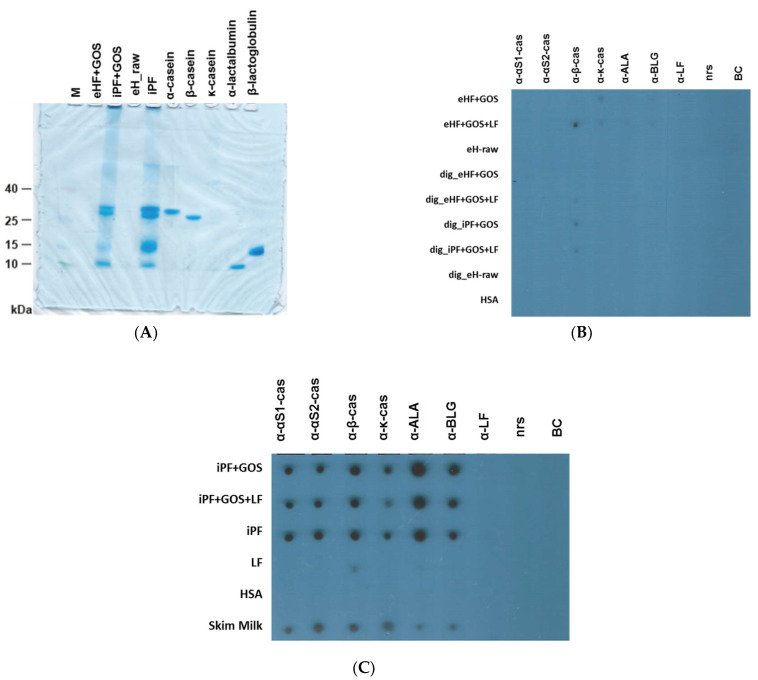
Analysis of materials by SDS-PAGE and dot blotting. (**A**) Infant formulas and milk allergens as indicated on the top (details described in Table 2) were separated by SDS-PAGE and stained with Coomassie Brilliant Blue. A molecular weight marker (lane M) was included, and molecular weights are indicated on the left margin. (**B**,**C**) Infant formulas, materials and proteins as indicated on the left margin were dotted onto nitrocellulose and then probed with rabbit antisera raised against milk allergens (α-S1-cas, α-S2-cas, α-β-casein, α-κ-casein, α-lactalbumin, α-β-lactoglobulin, and α-lactoferrin), a normal rabbit serum (nrs), or only buffer (**B**,**C**). Bound rabbit antibodies were detected with ^125^I-labeled antibodies and visualized by autoradiography.

**Figure 2 nutrients-15-00111-f002:**
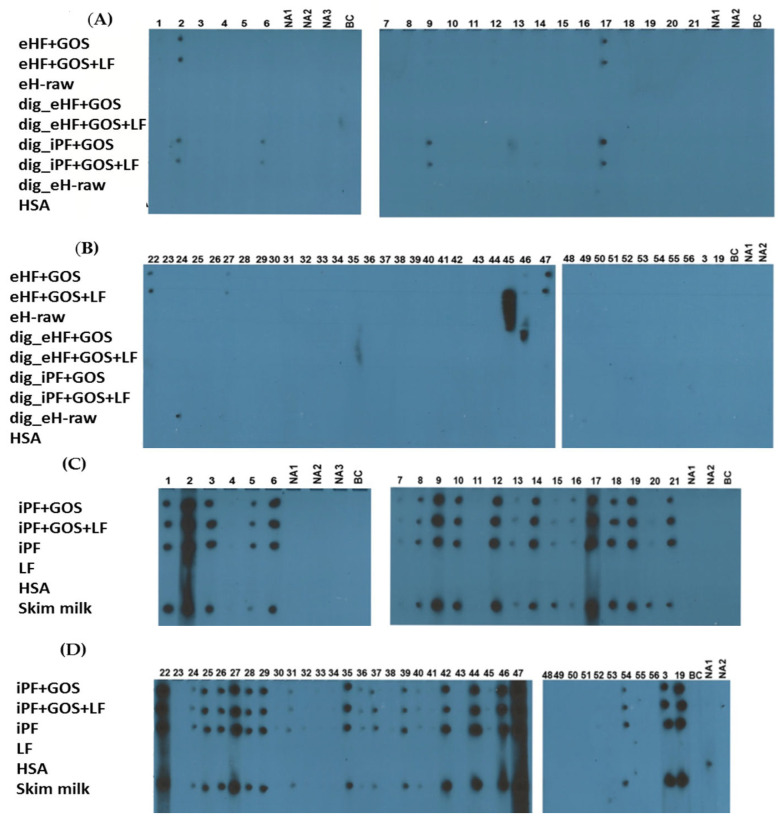
IgE reactivity to dot-blotted infant formulas or materials. (**A**–**D**) Dot-blotted infant formulas or materials (left; details described in Table 2) were exposed to sera from cow’s milk-allergic patients (1–56), sera from non-allergic subjects (NA1, NA2), or buffer alone (**B**,**C**). Bound IgE antibodies were detected with ^125^I-labeled anti-human IgE antibodies and visualized by autoradiography.

**Figure 3 nutrients-15-00111-f003:**
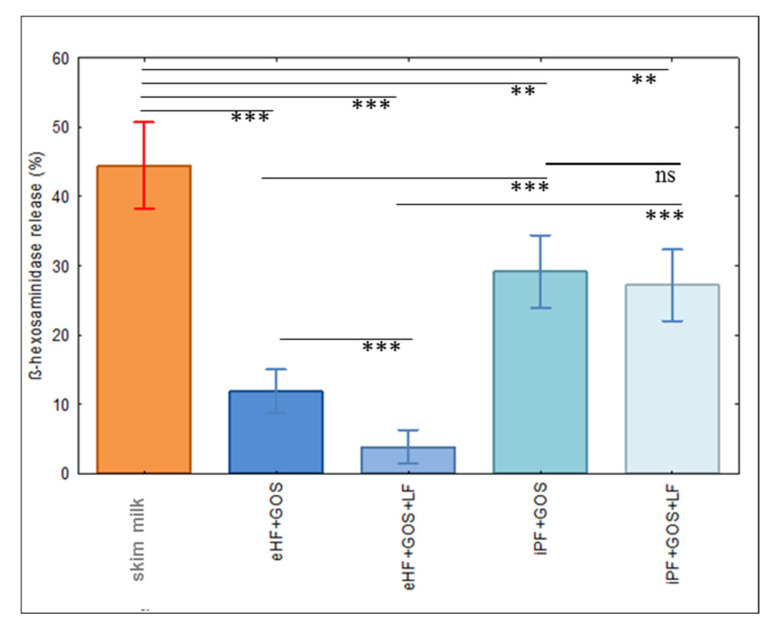
Specific basophil degranulation induced with milk products. Shown are the mean (n = 20) +/− SEM percentages of ß-hexosaminidase releases (*y*-axis) induced by a concentration of 10 ng/mL of different milk products or infant formula (*x*-axis; Table 2) from human FcεRI-expressing rat basophils that had been loaded with serum IgE from 20 different cow’s milk-allergic patients. Significant differences in mediator release induced by the milk products or infant formula are indicated. ** *p* ≤ 0.01, *** *p* ≤ 0.001, ns, not significant.

**Figure 4 nutrients-15-00111-f004:**
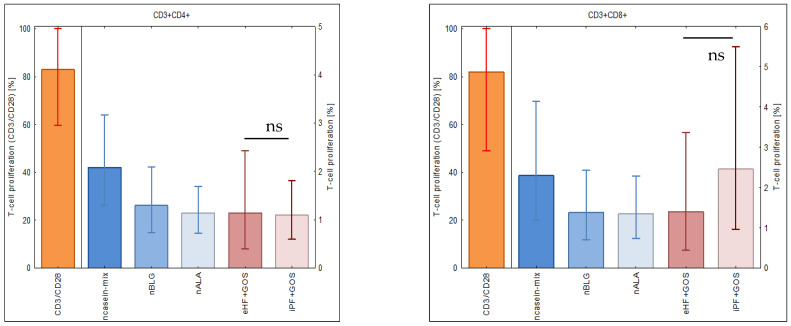
T cell proliferation specific for milk allergens, eHF + GOS and iPF + GOS. Shown are the back-transformed mean percentages +/− SEM of proliferated CD4^+^ (left part) and CD8^+^ T cells (right part) measured in PBMC samples from subjects 48–56 (Table 1) that had been stimulated with anti-CD3 and anti-CD28 (positive control), milk allergens or two milk products (*x*-axis; Table 2). Only eHF and iPF were compared statistically (ns, not significant).

**Figure 5 nutrients-15-00111-f005:**
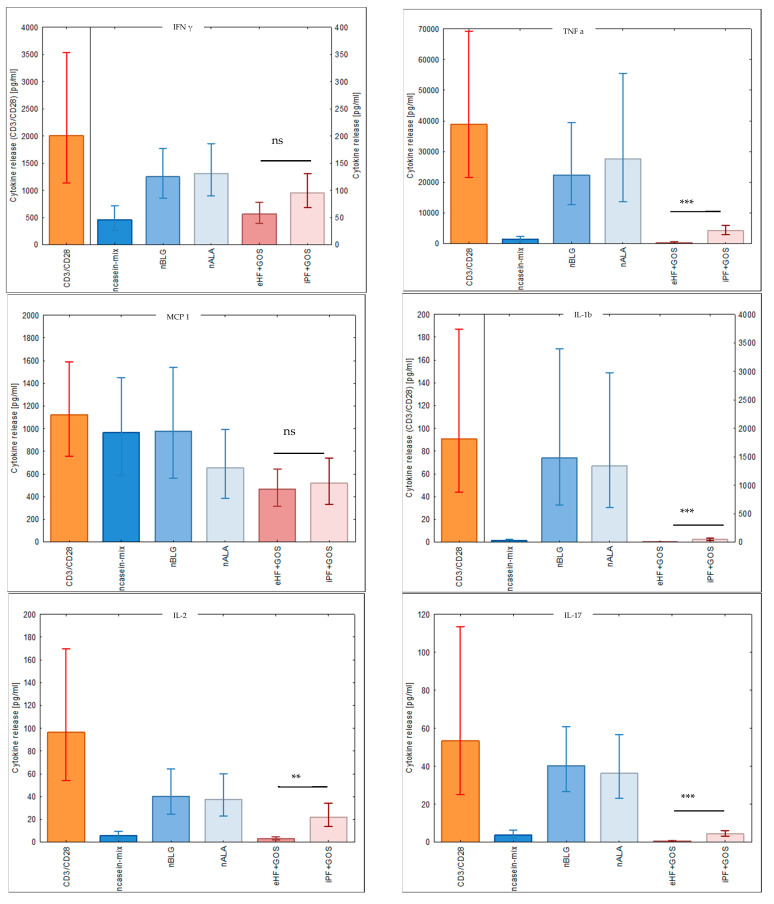
Cytokine responses specific for milk allergens and milk products. Shown are the back-transformed means +/− SEM of cytokines (as indicated on top of each figure) measured in supernatants of cultured PBMC samples from subjects 48–56 (Table 1) that had been stimulated with anti-CD3 and anti-CD28 (positive control), milk allergens, eHF + GOS and iPF + GOS (*x*-axis; Table 2). Only eHF and iPF were compared statistically (ns, not significant, ** *p* ≤ 0.01, *** *p* ≤ 0.001).

**Table 1 nutrients-15-00111-t001:** Demographic and clinical characteristics of allergic patients and of non-allergic individuals.

Number	Age	Sex	CMP-Related Clinical	SPT CMP	Allergy to Other Food Allergen
	Years	F/M	Symptoms		Sources
1	0.9	M	A, AD	pos	no
2	12.0	M	A, AD	pos	lentil
3	5.0	M	A, AD	pos	egg, wheat, walnut
4	2.4	F	A, AD	pos	egg
5	8.0	M	A	pos	egg
6	0.5	M	AD	pos	wheat, fish
7	4.0	M	AD	n.d.	peach, apricot
8	8.0	F	A, AD	n.d.	beef, egg
9	4.1	M	A, AD	n.d.	beef, pork, egg, wheat
10	1.9	F	AD	pos	egg, potato, wheat
11	3.4	M	AD	n.d.	egg
12	3.0	F	AD	n.d.	beef, egg
13	1.3	M	AD	n.d.	egg
14	6.1	M	A, AD	n.d.	egg
15	4.5	F	AD	pos	egg
16	1.6	M	AD	n.d.	no
17	8.0	F	A, AD	n.d.	no
18	10	F	AD	n.d.	egg, fruits, vegetables, cereals, fish, meat
19	6.0	M	AD	n.d.	fish, egg, meat, potato
20	1.1	M	A, AD	pos	wheat, fish
21	8.0	M	A, AD	pos	egg, walnut
22	2.9	F	A, GI	pos	no
23	2.9	F	A, GI	n.d.	egg
24	7.2	M	AD, GI	pos	peanut
25	3.7	M	A, AD, GI	n.d.	egg
26	5.0	F	A, AD, GI	pos	egg, nuts, caviar
27	2.9	F	A, AD, GI	n.d.	no
28	2.5	F	A, AD, GI	n.d.	no
29	4.1	M	AD, GI	n.d.	no
30	11.2	M	A, GI	pos	no
31	6.0	M	AD, GI	n.d.	egg
32	6.1	F	A, GI	pos	egg
33	4.2	M	A, AD, GI	n.d.	no
34	1.3	M	AD, GI	pos	egg, cereals, fish
35	10.6	M	A, AD, GI	pos	egg
36	7.3	F	AD, GI	n.d.	egg, oat
37	3.4	M	A, GI	n.d.	egg
38	1.4	M	A, AD, GI	n.d.	egg, cereals
39	5.6	M	AD, GI	n.d.	egg, wheat, caviar
40	6.6	M	AD, GI	n.d.	egg, wheat, caviar
41	1.8	M	A, AD, GI	n.d.	egg, wheat, walnut
42	4.8	M	A, AD, GI	n.d.	egg, walnut, peanut, soy
43	10.9	M	A, AD, GI	pos	no
44	3.2	M	A, AD, GI	n.d.	egg, wheat, nuts, soy
45	2.5	M	A, AD, GI	n.d.	egg
46	2.3	M	A, AD, GI	n.d.	egg
47	10.8	M	A, AD, GI	n.d.	peanut, soy, walnut
52	1.7	M	AD, GI	n.d.	egg, salmon, codfish
54	2.5	M	A	n.d.	egg, walnut
NCMA 48	1.7	M	no	n.d.	peanut
NCMA 49	11.3	M	no	n.d.	hazelnut, codfish
NCMA 50	11.1	M	no	n.d.	no
NCMA 51	8.2	M	no	n.d.	no
NCMA 53	1.5	M	no	n.d.	egg
NCMA 55	5.3	M	no	n.d.	peanut
NCMA 56	9.4	M	no	n.d.	peanut
NA1	3.0	F	no	neg	no
NA2	2.1	M	no	neg	no
NA3	0.7	F	no	neg	no

F, female; M, male; CMP, cow milk proteins; SPT, skin prick test (wheal area); A, anaphylaxis; AD, atopic dermatitis; GI, gastrointestinal symptoms; NCMA, non-cow’s milk allergic; NA, non-allergic; no, no symptoms to other food allergen sources; n.d., not done; pos, positive; neg, negative.

**Table 2 nutrients-15-00111-t002:** Description of investigated materials.

No.	Abbreviation	Description	BCA mg/mL	BCA Peptide mg/mL
1	eH_raw	Raw material: extensively hydrolyzed whey protein	n.a.	93.29
2	iPF	HiPP standard cow’s milk infant formula (HiPP Pre BIO, powder) w/o synbiotics	25.45	n.a.
3	LF	*Limosilactobacillus fermentum* CECT5716 (originally obtained from human milk)	2.26	n.a.
4	eHF + GOS	HiPP HA infant formula (HiPP Pre HA Combiotik^®^, liquid)	n.a.	20.92
5	eHF + GOS + LF	HiPP HA infant formula (HiPP Pre HA Combiotik^®^, liquid) + *L. ferm.* CECT5716	n.d.	n.d.
6	iPF + GOS	HiPP standard cow’s milk infant formula (HiPP Pre Bio Combiotik^®^, liquid)	22.64	n.a.
7	iPF + GOS + LF	HiPP standard cow’s milk infant formula (HiPP Pre Bio Combiotik^®^, liquid) + *L. ferm.* CECT5716	n.d.	n.d.
8	dig_eH_raw	In vitro-digested raw material: extensively hydrolyzed whey protein	n.a.	5.27
9	dig_eHF + GOS	In vitro-digested HiPP HA infant formula (HiPP Pre HA Combiotik^®^, liquid)	n.a.	6.30
10	dig_eHF + GOS + LF	In vitro-digested HiPP HA infant formula (HiPP Pre HA Combiotik^®^, liquid) + *L. ferm.* CECT5716	n.a.	8.94
11	dig_iPF + GOS	In vitro-digested HiPP standard cow’s milk infant formula (HiPP Pre Bio Combiotik^®^, liquid)	8.66	n.a.
12	dig_iPF + GOS + LF	In vitro-digested HiPP standard cow’s milk infant formula (HiPP Pre Bio Combiotik^®^, liquid) + *L. ferm.* CECT5716	8.97	n.a.
13	HSA	Human serum albumin (neg. ctl.)	1	n.a.
14	Skim milk powder	Commercial cow’s milk powder, Sigma Aldrich (pos. ctl.)	1	n.a.

eHF, extensively hydrolyzed formula; iPF, intact protein formula; GOS, Galactooligosaccharides; *L. ferm*, *L. fermentum* CECT5716; dig, digested (in-vitro infant SHIME^®^ model, Prodigest BV). HA infant formula is based on extensively hydrolyzed cow’s milk (whey) protein (eH_raw; Peptigen^®^ IF-3080, Arla Foods Ingredients). Intact protein formula is based on intact cow’s milk protein; BCA, bicinchoninic acid assay; n.a., not applicable; n.d., not done.

## Data Availability

Data will be made available by the authors upon reasonable request.

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
