# Peer review of "Extensively Hydrolyzed Hypoallergenic Infant Formula with Retained T Cell Reactivity"

_nutrients, 2022, doi:10.3390/nu15010111_

Round 1
Reviewer 1 Report
nutrients-1978614
The manuscript of R. Freidl et al. describes the in vitro studies evaluation of the immunoreactivity of formulas proposed for the prevention/treatment of milk protein allergy. In my opinion, it is a very significant and constantly topical issue.
The techniques used are consistent with the recommendations for this type of research, however, the planning of the research itself, and the selection of groups and variants for research, have considerable shortcomings. The manuscript itself was prepared carelessly and its appearance aroused a lot of frustration, so it was more difficult to focus on the content. In my opinion, many amendments and general supplementations are needed here to be able to accept the manuscript. However, the idea is significant.
Line 68: The conclusion and reference is has been discussed many times since then, and it has been questioned that the level of specific IgE correlates with the strength of the reaction. The position was taken that the sensitivity and strength of the reaction is a highly individual issue, and the presence of high levels of IgE antibodies, even specific, in many cases, may still be correlated with the supply and may do not give any symptoms.
Line 76- It has been confirmed significant role of IgG1 subclass in predating IgE epicutaneous sensitization to foods so it is should not be generalized that IgG is blocking IgE binding. I generally would recommend testing also IgG1- binding capacity of formulas
Line 82- What is more not only IgE-based tests are considered to be not reliable but also the provocation conducted in a clinical condition is the only reliable method of food allergy confirmation by many units in the world. For many years it is considered to be a gold standard.
Line 90- Although the EAACI recommends immunotherapy, it further confirms that the use of desensitization is a highly individual matter, emphasizes that there are no established uniform and fully effective protocols and that both the dose, the form of formula, the time of administration is an individual matter and the impermanence of the developed tolerance is still an unresolved issue.
Line 123- N-3 for control in a study where 59 allergic individuals is incomprehensible disproportion. Although testing 59 individuals in the control group does not seem necessary, I do not really understand the selection of the n-3 sample.
Line 123- I see 8 individuals like that (48, 49, 50, 51, 53, 55,56). The table is confusing and lot of mistakes.
Line 131- and not und
Line 150- A for bronchial asthma and A for Anaphylaxis??? So either you had a lot of anaphylactic patients or with astma. But there is another issue that you had no patients with one of that symptoms.
Line 150- What does it mean no symptoms – no abbreviation.
Line 153- I would suggest separately describe allergens separately infant formulas –especially since you use an addition of bacterial culture which demands a detailed description of the culture preparation and handling, I would also suggest describing the part of digestion due to the SHIME is a really interesting model that requires practice and experience and also adjusting parameters to applied material. I think that the methodological description, due to the lack of journal limitation in terms of words number and due to the prestige of the journal, requires a better and more complete methodological description.
Line 154-156 catalog numbers and distributors of ingredients are missing
Line 160- Where is that bacteria from? What was the strain is that the one that you wrote under the table 2 (CECT5716) ?
Line 160- what does it mean freshly, alived, cultured, passaged??? How many times, provide conditions anything?
Line 161- Why such inoculum? 1.5x10^6. Was it cultured later, or just digested? What was the reason of using such combinations for that formula?
Table 2- Not transparent- to many abbreviations no order. I would start from the row 5, 6,7 and later combine it.
Line 175-179 – from the table explanation we can find out you applied Shime infant. Definitely, you need to add a separate section of methodology.
Line 179- what is the BCA for. No description no explanation. It appears first time in the table.
Line 199- why that sera were used (20) by why exactly that 20.
Line 201- Supplementary Materials Figure 1A (not A1). Supplementary data should be in a separate file. This is not prepared according to the standards of the journal.
Line 206- Please provide the equation.
Line 222- why did you use 50 µM beta-mercaptoethanol for medium
Line 223- double bracket ))
Line 228- Please provide the equation.
Line 245-262- I can understand why you applied the Generalized Estimating Equations (GEE) model with but I would also try to perform kinda PCA analysis. It is also not clear what test did you use when you mention ‘For the specific 255 basophil degranulation all formulas were compared against skin milk, and for eHF as 256 well as iPF formulas with and without LF were compared and additionally eHF vs iPF 257 for both, galactooligosaccharide (GOS) and GOS+LF, by linear contrast with Bonferroni 258 correction.’ I also do not know why you applied the Bonferroni correction here. Please provide how you counted that correction because it seems it eliminated significant differences in the presented cytokine analyzes.
Line 256- skim milk not skin ( safer to use skimmed)
Line 267-269- purified natural CM allergens – where is that from. Did you buy it or you purified it by yourself? If you made it please provide a reference or describe how you made it.
Figure 1A- the quality of the gel is weak. you can hardly see the molecular mass marker, which makes it impossible to assess the size of the bands. It is also curved. Moreover, the k-casein is not visible and the alpha (should be approximately 14kDA) and beta-lactoglobulin (about 18kDA) masses seem too low as seen in the picture.
Line 275- do not start the sentence from next
Line 276- in all manuscript the application of antisera is confusing- because it seems it is direct detection not sandwich method with primary and secondary detection antibodies. Antisera are usually secondary antibodies against specific primary (like anti-human IgE)
Line 276- how did you prepare rabbit sera against particular proteins? Please provide the protocol of rabbit sensitization and purification of rabbits. Or give a reference to your previous study.
Line 279-kap-pa- casein,
Line 279- it is not consisten once you use alpha beta and sometimes symbols α,β. Unify that
Line 282- In all manuscript the application of statement ‘intact protein formulas’ is really confusing because it is protein formula and it is mentioned that it is digested so it is not anymore intact> I would recommend to change the word intact to the one consistent with table 2 provided means standard milk formula
Line 299 and 301- what does it mean normal rabbit serum ???
Line 301- which buffer control –precise
Line 310- how do you estimate the strength to grade it was strong. Is that arbitrary or you have any standard curve or something to calibrate
Line 319-how is that possible if you did not perform fermentation ? You mentioned you added fresh culture and digested it. The results suggest that the fermentation went spontaneously at the time of digestion. It could happen in shime . That suggests that to test the effect of the formula with the strain you should have also the formula with disintegrated cells. That experimental design have some missing points.
Line 361- In the term of obtained result of basophil test I would suggest also adding the parallel testing with digested bacterial cells
Figure 5- it is hard to believe there were no significant stated.
Figure 5- IL1B- the lrigh axis is to 4000 ?? or to 40? It is not visible and confusing.
Line 556-557- why Only eHF and iPF were compared statistically. Why you did not perform multicomparison test here. Why you did not perform the analysis for the previously mentioned formulas with bacterial strain that seems promising too.
Line 599- I would recommend adding 2 formulas to the experimental design (one as a pure strain and one milk-based formula with a disintegrated strain) to conclude the properties of bacterial strain.
Line 619-623- I am not really convinced to put such a conclusion. Especially that the sensitivity is an individual matter of every single case and I would be never so brave to conclude like that. Especially that you mentioned that IgE-mediated reaction is only one possible mechanism and that serum-based tests are not so reliable and trustworthy.
General comments:
1) The statement deep hydrolysis is not confirmed, no results of the referenced method of the degree of hydrolysis measurement.
2) I would also recommend performing LCMS/ MS verification of formulas peptidome to check how deep the hydrolysis is performed. Even 6 aa structures can induce the immune system and 9 aa is enough to fit and activate MHC receptors.
3) No matter what the producer declares the good practice is to start the study with the chemical analysis of the composition of the material.
Author Response
Please, see the attachment.

Reviewer 2 Report
Authors have made an attempt to study the T cell reactivity of hydrolyzed hypoallergenic infant formula in Cow’s Milk Allergic individuals. They have used an approach of stimulating PBMCs or sera with Said formula and tracking T cell reactivity and Treg activity. The results are interesting and look promising from the data.
The article needs Major reconstruction of format and data representation. I have major concerns with the review before it can be ready for publication. My feedback is outlined below.
Major Concerns:
1. Staining data lacks mention of investigation TRegs activity though its discussed throughout the paper.(I found it in supplement)
2. Any reason why gating strategy not mentioned in preliminary results?
3. No mention of Hydrolyzed formula contents and method used to manufacture it and the source
4. Why is Basophilo activation test perfrmed using Rat Basophil Leukamia cells instead of actual whole blood from subjects?
5. How is Basophil degranulation measured?
6. What was the duration of stimulation for proliferation assays
7. Line no 199: Overnight is approximately how many hours?
Author Response
Point-by-point response
Manuscript ID nutrients-1978614
Extensively hydrolyzed hypoallergenic infant formula with retained T cell reactivity
Reviewer 1
The manuscript of R. Freidl et al. describes the in vitro studies evaluation of the immunoreactivity of formulas proposed for the prevention/treatment of milk protein allergy. In my opinion, it is a very significant and constantly topical issue.
The techniques used are consistent with the recommendations for this type of research, however, the planning of the research itself, and the selection of groups and variants for research, have considerable shortcomings. The manuscript itself was prepared carelessly and its appearance aroused a lot of frustration, so it was more difficult to focus on the content. In my opinion, many amendments and general supplementations are needed here to be able to accept the manuscript. However, the idea is significant.
Reply: We thank the reviewer for the appreciating the topic of our work and are grateful for the suggestions for improvement. We have revised the manuscript as requested by the reviewer.
Line 68: The conclusion and reference is has been discussed many times since then, and it has been questioned that the level of specific IgE correlates with the strength of the reaction. The position was taken that the sensitivity and strength of the reaction is a highly individual issue, and the presence of high levels of IgE antibodies, even specific, in many cases, may still be correlated with the supply and may do not give any symptoms.
Reply: We completely agree with the reviewer that high allergen-specific IgE levels are at best a hint for high clinical sensitivity but are not always associated with severe symptoms. This is the reason why we have performed our analysis with sera from clinically well characterized patients and included also basophil activation testing. We have revised the manuscript to state explicitly that we provide only results from in vitro experiments and that in vivo exposure studies for the infant formula will be necessary to confirm the reduction of allergenic activity. (see lines 679-687 of revised manuscript).
Line 76- It has been confirmed significant role of IgG1 subclass in predating IgE epicutaneous sensitization to foods so it is should not be generalized that IgG is blocking IgE binding. I generally would recommend testing also IgG1- binding capacity of formulas.
Reply: We thank the reviewer for this comment but international guidelines recommend against testing of IgG reactivity specific for food allergens (see: Testing for IgG4 against foods is not recommended as a diagnostic tool: EAACI Task Force Report Steven O Stapel 1 , R Asero, B K Ballmer-Weber, E F Knol, S Strobel, S Vieths, J Kleine-Tebbe, EAACI Task Force, Allergy, 2008 Jul;63(7):793-6. doi: 10.1111/j.1398-9995.2008.01705.x. Epub 2008 May 16.). Nevertheless we have mentioned the consideration to test also IgG1 binding of formulas when their clinical relevance has been proven (see lines 683-687).
Line 82- What is more not only IgE-based tests are considered to be not reliable but also the provocation conducted in a clinical condition is the only reliable method of food allergy confirmation by many units in the world. For many years it is considered to be a gold standard.
Reply: We agree with the reviewer that the provocation test is a gold standard for confirming food allergy. Accordingly we have used sera from clinically well characterized patients for the evaluation of the infant formula by IgE testing and the more relevant basophil activation testing which reflects the induction of immediate inflammation. We have mentioned this consideration in the revised manuscript (see lines 679-687).
Line 90- Although the EAACI recommends immunotherapy, it further confirms that the use of desensitization is a highly individual matter, emphasizes that there are no established uniform and fully effective protocols and that both the dose, the form of formula, the time of administration is an individual matter and the impermanence of the developed tolerance is still an unresolved issue.
Reply: We fully agree with the reviewer and refined the statements regarding immunotherapy in our manuscript (see lines 97-99).
Line 123- N-3 for control in a study where 59 allergic individuals is incomprehensible disproportion. Although testing 59 individuals in the control group does not seem necessary, I do not really understand the selection of the n-3 sample.
Line 123- I see 8 individuals like that (48, 49, 50, 51, 53, 55,56). The table is confusing and lot of mistakes.
Reply: We thank reviewer for the careful checking of the table. We have revised the table 1 and indicated the 7 patients who are allergic but not to cow´s milk (NCMA 48, 49, 50, 51, 53, 55 and 56) and the three non-allergic subjects (NA1-3) in the end as control subjects. We thus have in total 49 cow´s milk allergic subjects and ten subjects without cow´s milk allergy which seems appropriate. Furthermore we corrected the legend for Table 1.
Line 131- and not und
Reply: The typo was corrected (line141)
Line 150- A for bronchial asthma and A for Anaphylaxis??? So either you had a lot of anaphylactic patients or with astma. But there is another issue that you had no patients with one of that symptoms.
Reply: A stands for anaphylaxis defined by Sampson (see reference 19: Sampson, H.A.; Muñoz-Furlong, A.; Campbell, R.L.; Adkinson, N.F.; Bock, S.A.; Branum, A.; Brown, S.G.; Camargo, C.A.; Cydulka, R.; Galli, S.J.; et al. Second symposium on the definition and 15 management of ana-phylaxis: summary report--Second National Institute of Allergy and 16 Infectious Dis-ease/Food Allergy and Anaphylaxis Network symposium. J. Allergy Clin. Immunol. 2006, 117, 391-397.) We have corrected this (see revised Table 1). There were no cow´s milk allergic patients suffering from cow´s milk induced bronchial asthma.
Line 150- What does it mean no symptoms – no abbreviation.
Reply: The abbreviation “no” meant “no allergy to other food allergen sources”. It is explained in the revised footnote of Table 1.
Line 153- I would suggest separately describe allergens separately infant formulas –especially since you use an addition of bacterial culture which demands a detailed description of the culture preparation and handling, I would also suggest describing the part of digestion due to the SHIME is a really interesting model that requires practice and experience and also adjusting parameters to applied material. I think that the methodological description, due to the lack of journal limitation in terms of words number and due to the prestige of the journal, requires a better and more complete methodological description.
Reply: We agree with reviewer, that the SHIME is an interesting model. In fact, the digestion has been performed as a customized professional service by Prodigest BV (Gent, Belgium, https://prodigest.eu/). This information has been added to the revised methods section (lines 185-186).
Line 154-156 catalog numbers and distributors of ingredients are missing#
Reply: Purified natural cow’s milk allergens, skim milk powder and human serum albumin (HSA) were purchased from Sigma Aldrich (St. Louis, Missouri). The catalogue numbers have now been included in the revised methods section (lines 164-165). All tested formulas and raw materials were provided by HiPP GmbH & Co. Vertrieb KG (Pfaffenhofen, Germany) and Arla Food Ingredients (Viby, Denmark), respectively.
Line 160- Where is that bacteria from? What was the strain is that the one that you wrote under the table 2 (CECT5716) ?
Reply: The bacteria were originally isolated from human breast milk. We just used one strain for all experiments (i.e., L. fermentum CECT5716) (HiPP GmbH & Co. Vertrieb KG (Pfaffenhofen, Germany). This information was included in the revised methods (see lines 167-168).
Line 160- what does it mean freshly, alived, cultured, passaged??? How many times, provide conditions anything?
Reply: L. fermentum CECT5716 was obtained as a freeze-dried stock and then just defrosted before bacteria were added. The text was corrected accordingly to make clear that bacteria were no further cultured or passaged (see lines 181-184 of revised methods).
Line 161- Why such inoculum? 1.5x10^6. Was it cultured later, or just digested? What was the reason of using such combinations for that formula?
Reply: This concentration of L.fermentum CECT5716 mentioned by the reviewer is already used in powdered Hipp Infant formula which is already on the market. Furthermore, similar concentrations have also been used in a clinical trial (ClinicalTrials.gov Identifier: NCT02221687). To investigate the effects of this defined concentration L.fermentum CECT5716 freeze-dried bacteria were freshly to sterile liquid HA formula. We have added this information to the revised methods section (see lines 181-184).
Table 2- Not transparent- to many abbreviations no order. I would start from the row 5, 6,7 and later combine it.
Reply: We have modified the order in Table 2 as requested by the reviewer (see new Table 2).
Line 175-179 – from the table explanation we can find out you applied Shime infant. Definitely, you need to add a separate section of methodology.
Reply: As mentioned above, the digestion has been performed as a customized professional service by Prodigest BV (Gent, Belgium, https://prodigest.eu/). This information has been added to the revised methods section (lines 185-186).
Line 179- what is the BCA for. No description no explanation. It appears first time in the table.
Reply: BCA, the bicinchoninic acid assay is a biochemical assays for determining the concentration of proteins and peptides. An explanation has been added to the revised methods section (see lines 195-198).
Line 199- why that sera were used (20) by why exactly that 20.
Reply: Sera from a subset of 20 patients were tested due to limited volume of serum samples required to perform the basophil activation experiments.
Line 201- Supplementary Materials Figure 1A (not A1). Supplementary data should be in a separate file. This is not prepared according to the standards of the journal.
Reply: We thank the reviewer for this comment. In order to indicate that both figures belong to the supplementary file we designated the Figures now Figure S1 and Figures S2. The designation Figure 1A would have been misleading because it would have indicated that the Figure is part of the main manuscript. We apologize for the confusion.
Line 206- Please provide the equation.
Reply: The percentage of beta-hexosaminidase release was calculated taking lysis of cells with Triton-X 100 as 100% release reference. The methods section was revised to explain this (see lines 243-252).
Line 222- why did you use 50 µM beta-mercaptoethanol for medium
Reply: Beta-mercaptoethanol is used in cell culture to prevent free radical build-up.
Line 223- double bracket ))
Reply: corrected (see line 271-272)
Line 228- Please provide the equation.
Reply: We apologize for our mistake. For each of the 20 patients cultivation with the different antigens was done in triplicate. For each of the triplicates the median of the medium only wells was subtracted. For further statistical analysis, the mean of the three medium corrected proliferation values was used. (see lines 276-279)
Line 245-262- I can understand why you applied the Generalized Estimating Equations (GEE) model with but I would also try to perform kinda PCA analysis. It is also not clear what test did you use when you mention ‘For the specific 255 basophil degranulation all formulas were compared against skin milk, and for eHF as 256 well as iPF formulas with and without LF were compared and additionally eHF vs iPF 257 for both, galactooligosaccharides (GOS) and GOS+LF, by linear contrast with Bonferroni 258 correction.’ I also do not know why you applied the Bonferroni correction here. Please provide how you counted that correction because it seems it eliminated significant differences in the presented cytokine analyzes.
Reply: A PCA was not possible because the number of variables was in the range of the number of patients. The hypothesis test applied was Wald's chi-square test (we added this information in the revised version, lines 293-311). Correction was determined by the number of comparisons that are not orthogonal. We agree that this rather leads to conservative interpretation of findings.
Line 256- skim milk not skin ( safer to use skimmed)
Reply: Corrected (see line 305). The most suitable term is skim.
Line 267-269- purified natural CM allergens – where is that from. Did you buy it or you purified it by yourself? If you made it please provide a reference or describe how you made it.
Reply: As mentioned in the revised methods section (see lines 164-165), the purified natural CM allergens were purchased from Sigma Aldrich.
Figure 1A- the quality of the gel is weak. you can hardly see the molecular mass marker, which makes it impossible to assess the size of the bands. It is also curved. Moreover, the k-casein is not visible and the alpha (should be approximately 14kDA) and beta-lactoglobulin (about 18kDA) masses seem too low as seen in the picture.
Reply: We agree with the reviewer that the gel shows a smiling effect but the alpha lactalbumin and ß-lactoglobulin bands correspond well with the masses of the proteins in the infant formula IPF. The k-casein band is indeed weak. This can be explained that despite the fact that similar amounts of protein have been analyzed the formation of higher molecular weight oligomers had occurred. However, the most important result can be well seen in the gel. Extensively hydrolysed whey protein with (eHF+GOS) and without Galactooligosaccharides (lane eH_raw) does not contain intact proteins.
Line 275- do not start the sentence from next
Reply: Corrected (see line 325)
Line 276- in all manuscript the application of antisera is confusing- because it seems it is direct detection not sandwich method with primary and secondary detection antibodies. Antisera are usually secondary antibodies against specific primary (like anti-human IgE)
Reply: We thank the reviewer for this comment and have revised the methods section to clarify that the rabbit antisera raised against purified cow´s milk allergens were detected with secondary, 125Iodine-labelled anti-rabbit antibodies (see lines 214-216).
Line 276- how did you prepare rabbit sera against particular proteins? Please provide the protocol of rabbit sensitization and purification of rabbits. Or give a reference to your previous study.
Reply: The preparation of allergen-specific antisera was previously described by Hochwallner, et al. Microarray and allergenic activity assessment of milk allergens. Clin. Exp. Allergy 2010, 40, 1809-1818. doi: 10.1111/j.1365-2222.2010.03602.x. We have referenced this in the revised methods section (line 215).
Line 279-kap-pa- casein,
Reply: We are sorry for this typo which was due to the formatting program. It was corrected, see line 329.
Line 279- it is not consisten once you use alpha beta and sometimes symbols α,β. Unify that
Reply: We thank the reviewer for drawing our attention to this inconsistency. We have corrected it accordingly to “α-casein, β-casein, κ-casein, α-lactalbumin, β-lactoglobulin” throughout the manuscript (lines 327-329).
Line 282- In all manuscript the application of statement ‘intact protein formulas’ is really confusing because it is protein formula and it is mentioned that it is digested so it is not anymore intact> I would recommend to change the word intact to the one consistent with table 2 provided means standard milk formula
Reply: We thank the reviewer for this comment. In fact, the term “intact protein formula (iPF)” is used for a formula which was not hydrolysed. Indeed, we have prepared digested forms which are consistently named dig_iPF. In order to avoid confusion we have explained this definition (see line 186).
Line 299 and 301- what does it mean normal rabbit serum ???
Reply: With normal rabbit serum, serum from a non-vaccinated rabbit is meant. This was explained in the revised methods (see lines 188-190).
Line 301- which buffer control –precise
Reply: We used phosphate buffered saline with Tween (i.e., PBST). This was explained in the revised methods (see lines 222-224).
Line 310- how do you estimate the strength to grade it was strong. Is that arbitrary or you have any standard curve or something to calibrate
Reply: We thank the reviewer for the useful comment. The signals were visualized by autoradiography using Kodak XOMAT films with intensifying screens (Kodak, Vienna, Austria) at 80°C for the same time to allow a comparison. The intensity of the signals were then compared with each other and arbitrarily put into categories. This was explained in the revised methods section (see lines 226-228).
Line 319-how is that possible if you did not perform fermentation ? You mentioned you added fresh culture and digested it. The results suggest that the fermentation went spontaneously at the time of digestion. It could happen in shime . That suggests that to test the effect of the formula with the strain you should have also the formula with disintegrated cells. That experimental design have some missing points.
Reply: It seems that the reviewer refers to the following sentence in line 319: “Of note, patients 22 and 27 showed residual IgE reactivity to extensively hydrolyzed formulas but not to digested intact formulas whereas patient 9 show residual IgE reactivity to digested intact formulas but not to extensively hydrolyzed formulas”
This result can be explained by the fact that the cow´s milk-specific IgE epitopes recognized by patients 22 and 27 and those recognized by patient 9 are different. IgE epitopes recognized by patients 22 and 27 are destroyed by digestion whereas IgE epitopes recognized by patients 9 are sensitive to hydrolysis. Since IgE reactivity was the same regardless if L.fermentum was added or not (eHF+GOS, eHF+GOS+LF, dig_iPF+GOS, dig_iPF+GOS+LF) it did not depend on the strain.
Line 361- In the term of obtained result of basophil test I would suggest also adding the parallel testing with digested bacterial cells
Reply: We thank the reviewer for this suggestion but this experiment cannot be performed because sera needed for further basophil experiments have been used up. However, this experiment would also not provide additional information over the IgE binding data in Figure 2 which show that none of the sera showed IgE reactivity to LF. Accordingly one cannot expect IgE-dependent basophil activation by LF.
Figure 5- it is hard to believe there were no significant stated.
Reply: As requested above by the reviewer we have provided additional information regarding the statistical information. Only significant differences between eHF+GOS and iPF+GOS were analyzed. Results obtained for purified proteins were not considered because the goal was to compare eHF+GOS and iPF+GOS.
Figure 5- IL1B- the lrigh axis is to 4000 ?? or to 40? It is not visible and confusing.
Reply: Regarding IL-1ß the right y-axis valid for purified proteins as well as for eHF+GOS and iPF+GOS goes up to 4000.
Line 556-557- why Only eHF and iPF were compared statistically. Why you did not perform multicomparison test here. Why you did not perform the analysis for the previously mentioned formulas with bacterial strain that seems promising too.
Reply: We thank the reviewer for the thoughtful comments. Multiple testing would have required corrections and the key question of the paper was to compare eHF+GOS and iPF+GOS regarding their ability to induce T cell responses and cytokine responses. We therefore focused on this key question in the T cell and cytokine experiments. Furthermore, we were limited regarding the number of cells which we could obtain form the patients.
Line 599- I would recommend adding 2 formulas to the experimental design (one as a pure strain and one milk-based formula with a disintegrated strain) to conclude the properties of bacterial strain.
Reply: We thank reviewer for this comment but we were limited regarding the volumes of sera. We therefore focused on eHF+GOS+LF which corresponds to the final product intended for consumption. As pointed out above, testing of the strain alone by IgE-dependent basophil activation would not have provided information because the strain itself did not show any IgE reactivity (see Figure 2, LF).
Line 619-623- I am not really convinced to put such a conclusion. Especially that the sensitivity is an individual matter of every single case and I would be never so brave to conclude like that. Especially that you mentioned that IgE-mediated reaction is only one possible mechanism and that serum-based tests are not so reliable and trustworthy.
Reply: The hydrolysate was tested in a HiPP-RCT in 2021 (https://clinicaltrials.gov/ct2/show/NCT04736082). The study results were submitted to EFSA and the hydrolysate was approved to be safe (EFSA opinion: https://www.efsa.europa.eu/de/efsajournal/pub/7141 ). Nevertheless, we agree with the reviewer to be more cautious and modified our statement to say that this may be considered if further clinical studies confirm the presented preclinical data (see lines 690-696).
General comments:
- The statement deep hydrolysis is not confirmed, no results of the referenced method of the degree of hydrolysis measurement.
Reply: Following the reviewers request we provided information in the revised methods (see lines 171-180) regarding the hydrolysis process and how the degree of hydrolysis was determined.
“HiPP HA infant formula is manufactured from Peptigen® IF-3080 (supplied by Arla Foods Ingredients, Videbæk, Denmark). Peptigen® IF-3080 is an extensive whey protein hydrolysate suitable as a sole protein source in infant formula. It consists of short-chained peptides, obtained by a controlled enzymatic degradation of whey proteins. Hydrolysis is performed with food grade enzymes which were heat inactivated upon termination of hydrolysis. Subsequently, the hydrolysate is filtered by ultrafiltration in order to remove larger peptides and aggregates hereof. Peptigen® IF-3080 has a degree of hydrolysis of up to 30%. The degree of hydrolysis is defined as the percentage of peptide bonds cleaved by hydrolysis and determined according to Adler-Nissen and Nielsen et al. (Adler-Nissen, J. Determination of the degree of hydrolysis of food protein hydrolysates by trinitrobenzenesulfonic acid. J. Agric. Food Chem. 27, 1256-1262, 1979. Nielsen, P.M., Petersen, D. & Dambmann, C. Improved method for determining food protein degree of hydrolysis. J. Food Sci. 66, 642-646, 2001).”
- I would also recommend performing LCMS/ MS verification of formulas peptidome to check how deep the hydrolysis is performed. Even 6 aa structures can induce the immune system and 9 aa is enough to fit and activate MHC receptors.
Reply: As indicated in the reply above the degree of hydrolysis was up to 30%. The Coomassie-stained gel in Figure 1 shows that eHF+GOS and eH_raw do not contain any visible protein bands below 10 kDa. Importantly and eHF+GOS induced specific proliferation of CD4+ and CD8+ T cells and thus contain peptides which are in the size of 9aa or long enough to fit and activate MHC class I and MHC class II as indicated by the reviewer. We have added this consideration to the revised discussion (lines 665-670).
3) No matter what the producer declares the good practice is to start the study with the chemical analysis of the composition of the material. “No mention of Hydrolyzed formula contents and method used to manufacture it and the source”
Reply: Following the reviewers request we provided information in the revised methods (see lines 171-180) regarding the hydrolysis process and how the degree of hydrolysis was determined. “HiPP HA infant formula is manufactured from Peptigen® IF-3080 (supplied by Arla Foods Ingredients, Videbæk, Denmark). Peptigen® IF-3080 is an extensive whey protein hydrolysate suitable as a sole protein source in infant formula. It consists of short-chained peptides, obtained by a controlled enzymatic degradation of whey proteins. Hydrolysis is performed with food grade enzymes which were heat inactivated upon termination of hydrolysis. Subsequently, the hydrolysate is filtered by ultrafiltration in order to remove larger peptides and aggregates hereof. Peptigen® IF-3080 has a degree of hydrolysis of up to 30%. The degree of hydrolysis is defined as the percentage of peptide bonds cleaved by hydrolysis and determined according to Adler-Nissen and Nielsen et al. (Adler-Nissen, J. Determination of the degree of hydrolysis of food protein hydrolysates by trinitrobenzenesulfonic acid. J. Agric. Food Chem. 27, 1256-1262, 1979. Nielsen, P.M., Petersen, D. & Dambmann, C. Improved method for determining food protein degree of hydrolysis. J. Food Sci. 66, 642-646, 2001).”
The degree of hydrolysis was up to 30%. The Coomassie-stained gel in Figure 1 shows that eHF+GOS and eH_raw do not contain any visible protein bands below 10 kDa. Importantly, eHF+GOS induced specific proliferation of CD4+ and CD8+ T cells and thus contain peptides which are in the size of 9aa or long enough to fit and activate MHC class I and MHC class II as indicated by the reviewer. Furthermore, the hydrolysate was tested in a HiPP-RCT in 2021 (https://clinicaltrials.gov/ct2/show/NCT04736082) for safety. The study results were submitted to EFSA and the hydrolysate was approved to be safe (EFSA opinion: https://www.efsa.europa.eu/de/efsajournal/pub/7141 ).
We have added this information to the revised discussion (lines 665-670, 690-692).

Round 2
Reviewer 1 Report
The manuscript nutrients-1978614.
Article gained in readability and quality after taking into account the reviews. In the opinion of the reviewer, it may be admitted to the publication process. I still have doubts about the role of the sponsor, but unfortunately such application tests carry the risk of such doubts. In the opinion of the reviewer, commercialization research should be published in industry, not scientific journals. However In my opinion, the conclusions are consistent with the evidence and arguments presented here address the main aim.
Reviewer 2 Report
The authors have made necessary changes requested in the review report. There are minor edits in Figure no 4 and 5 graph Y axis legend. Some graphs mention cytokine release in CD3/CD28.
After these minor edits, manuscript can be approved in my opinion.